# Machine Learning Derived Lifting Techniques and Pain Self-Efficacy in People with Chronic Low Back Pain

**DOI:** 10.3390/s22176694

**Published:** 2022-09-04

**Authors:** Trung C. Phan, Adrian Pranata, Joshua Farragher, Adam Bryant, Hung T. Nguyen, Rifai Chai

**Affiliations:** 1School of Science, Computing and Engineering Technologies, Swinburne University of Technology, Hawthorn, VIC 3122, Australia; 2School of Health Sciences, Swinburne University of Technology, Hawthorn, VIC 3122, Australia; 3School of Kinesiology, Shanghai University of Sports, Shanghai 200438, China; 4Centre for Health, Exercise and Sports Medicine, Department of Physiotherapy, The University of Melbourne, Melbourne, VIC 3010, Australia

**Keywords:** low back pain, lifting technique, camera system, ward clustering method, K-means clustering method, ensemble clustering method, Bayesian neural network, pain self-efficacy questionnaire

## Abstract

This paper proposes an innovative methodology for finding how many lifting techniques people with chronic low back pain (CLBP) can demonstrate with camera data collected from 115 participants. The system employs a feature extraction algorithm to calculate the knee, trunk and hip range of motion in the sagittal plane, Ward’s method, a combination of K-means and Ensemble clustering method for classification algorithm, and Bayesian neural network to validate the result of Ward’s method and the combination of K-means and Ensemble clustering method. The classification results and effect size show that Ward clustering is the optimal method where precision and recall percentages of all clusters are above 90, and the overall accuracy of the Bayesian Neural Network is 97.9%. The statistical analysis reported a significant difference in the range of motion of the knee, hip and trunk between each cluster, *F* (9, 1136) = 195.67, *p* < 0.0001. The results of this study suggest that there are four different lifting techniques in people with CLBP. Additionally, the results show that even though the clusters demonstrated similar pain levels, one of the clusters, which uses the least amount of trunk and the most knee movement, demonstrates the lowest pain self-efficacy.

## 1. Introduction

Chronic low back pain (CLBP) is a multifactorial condition that is the leading cause of activity limitations and work absenteeism, affecting 540 million people worldwide [1]. Adaptation in trunk muscle control is commonly observed in people with CLBP, which is associated with changes in trunk muscle properties [2] and delayed reaction time in response to external perturbations [3]. These adaptations could be reflected in trunk and lower limb movement variability, especially during lifting [4]. 

Lifting is a complex activity that requires coordination of the lower limbs (e.g., hip and knee) and the trunk [4]. In simplistic terms, lifting techniques could be classified as a stoop lift (i.e., lifting with flexed back) or leg lift (i.e., lifting with hips and knees bent and back straight). Although lifting with the legs was traditionally considered to be a safer lifting technique, this has been disputed in several studies [5,6]. Lifting movements can vary considerably between individuals depending on factors such as hamstring tightness and movement speed—both of which have been demonstrated in people with CLBP or in risk factors for CLBP [7,8,9]. Therefore, dichotomous classification of lifting techniques may not be appropriate in people with CLBP. It is currently unknown how many different lifting techniques people with CLBP would demonstrate. This information may guide clinicians in identifying and individualizing target areas for rehabilitation for people with CLBP.

Moreover, it is well established that CLBP is associated with changes in psychosocial domain such as pain self-efficacy [10]. Pain self-efficacy is defined as the belief in one’s ability to perform painful or perceived painful tasks or movements to achieve a desirable outcome. Pain self-efficacy is typically measured using a Pain Self-Efficacy Questionnaire (PSEQ) [11]. In people with CLBP, low pain self-efficacy is associated with higher pain intensity, disability, and fear-avoidance beliefs [12,13,14]. Therefore, pain self-efficacy is an important attribute to be assessed in people with CLBP.

Recent technology using the computational intelligence technique for classification [15] may assist with the identification of different lifting techniques in people with CLBP. In principle, there are three main steps for activity recognition: (i) data capture by appropriate sensor; (ii) segmentation of the captured data and feature extraction; (iii) recognition of the activity using appropriate classification/identification techniques.

In classification, machine learning is known as one of the categories of artificial intelligence. In general, there are two types of machine learning: supervised and unsupervised. In supervised machine learning, once the data set has been labelled with each input, a pre-set correct output is assigned [16]. By contrast, unsupervised machine learning techniques utilize unlabelled data sets to identify patterns which will then be clustered into different groups [16]. In different medical and health applications, clustering algorithms have been applied to cluster patient records to identify a trend in health care [17,18], detect a set of co-expressed genes [19], categorize patients from medical records [20], and from the symptoms, find out patient subgroups [21]. It is unknown whether different movement patterns could be identified using unsupervised machine learning techniques or clustering algorithms in people with CLBP. 

Thus, this study aims to present an innovative methodology for identifying different lifting movement patterns in people with CLBP using unsupervised machine learning techniques and range of motion. Therefore, the main contribution of this paper is the novel application of unsupervised machine learning techniques for lifting movement pattern classification in the CLBP participants. We hypothesized that people with CLBP will lift utilizing various techniques when clustered using the trunk, hip and knee movement integration.

## 2. Materials and Methods

The components for the camera-based cluster classification system introduced in this paper are presented in Figure 1.

### 2.1. Participants

One-hundred and fifteen males and females (*n*_females_ = 57) with CLBP aged 25 to 60 years old with CLBP were recruited from a large physiotherapy clinic in Melbourne (VIC, Australia). This study was approved by the University of Melbourne Behavioural and Social Sciences Human Ethics Sub-Committee. Participants were included in the study if they had reported pain between the gluteal fold and the twelfth thoracic vertebra (T12) level, with or without leg pain that had persisted for >3 months. Participants were excluded from the study if they demonstrated overt neurological signs, such as muscle weakness and loss of lower limb reflexes, had had spine and lower limb surgery, had been diagnosed with active inflammatory conditions such as rheumatoid arthritis, had been diagnosed with cancer or did not comprehend written or verbal English. The participants in this study had not received any physiotherapy intervention during their assessment of lifting technique. All participants completed assessments of pain self-efficacy using the PSEQ [10].

### 2.2. Data Collection

The lifting task protocol has been published previously [4,22]. In summary, participants began the test standing upright, barefooted, with their arms by their sides. Participants were instructed to bend down and lift an 8 kg weight (i.e., an average weight of groceries [23]) placed between their feet with both hands from the ground up to the level of their abdomen. Participants were instructed to utilize a lifting technique of their choosing. Eight lifting trials were performed, with the first 2 trials serving as practice trials, hence excluded from data analysis. The sequence of consecutive actions during the lifting task is summarized and presented in Figure 2.

Kinematic data were collected using non-reflective markers placed on the participants’ skin to mark the head, trunk, pelvis, upper and lower limbs [22]. A 12-camera motion analysis system (Optitrack Flex 13, NaturalPoint, Corvallis, OR, USA) with 120 Hz sampling rate was utilized to provide the three-dimensional recording of anatomical landmarks. Kinematic data were grouped, named, cleaned and gap-filled using Optitrack Motive software (NaturalPoint, Corvallis, OR, USA). The data were then passed through to a custom-written analysis pipeline of Visual3D v5.01.6 (C-Motion, Inc., Germantown, MD, USA). Angular displacement and velocity data of different joints in all planes were derived using custom-written software (LabVIEW 2009, National Instruments).

### 2.3. Pre-Processing and Features Extraction

The angular displacement of the trunk, hip and knee joints during lifting were used for analysis and were inputted into the machine learning algorithm. 

A joint range of motion is chosen to simplify the complex data into efficient features. The joint range is calculated by taking the difference between maximum and minimum values of that joint’s angular displacement.
(1)Range of motion (ROM)=Max(θ)−Min(θ)
where Max(θ) is the maximum value of joint’s angular displacement, and Min(θ) is the minimum value of the joint’s angular displacement.

One perspective of inter-joint coordination in manual lifting is a distal-to-proximal pattern of extension of the knee, hip, and lumbar vertebral joints [24]. In addition, the movement of the knee, the hip and the lumbar take an important role in achieving the lifting task and generating different types of lifting techniques. From the provided data, the range of motion of the trunk, hip and knee in the sagittal plane are extracted for further analysis. A between-side average value was used for the knee and hip, as there was no statistically significant ROM differences between the left and right sides.

### 2.4. Partitional Clustering

Clustering is known as one of the common techniques which is used to generate homogeneous groups of objects [25]. From the provided data points, all the data points that are similar and closely resemble each other will be placed into the same cluster [26]. Partitional clustering is known as one of the most popular algorithm in clustering [27,28,29]. In partitional clustering, data points are separated into a predetermined number of clusters without a hierarchical structure. 

The K-means clustering algorithm is most commonly used as a partitional clustering [30]. In K-means clustering algorithm, h clusters are generated so that the distance between the points within its own clusters and centroids is less than the distance to centroids of other clusters. The algorithm’s operation begins with the selection of h points as the centroid. Following the selection of the h points, all points are allocated to the closest centroid, resulting in the formation of t clusters. The average of each cluster’s points will then be used to generate a centroid. These centroids make up the mean vector, with each field of the vector equalling each cluster centroid. A new centroid generates the new cluster as a result of this process. In the situation where the centroids remain unchanged, K-means clustering algorithm will be completed. K-means clustering has certain advantages such as less space and time complexity or optimal results accomplished for equal-sized globular and separated clusters [30]. However, K-means clustering algorithm shows sensitivity to outliers and noises. Moreover, a poor initial choice of centroid in partitioning process might produce increasingly poor result. In this study, K-means clustering algorithm was implemented using kmean function from Matlab. 

### 2.5. Ensemble Clustering

In recent years, clustering ensemble has widely been used to improve the robustness and quality of results from clustering algorithms [31,32,33,34]. In ensemble clustering, multiple results from different clustering algorithms are combined into final clusters without retrieving features or base algorithm information. In ensemble clustering, the only requirement is obtaining the base clustering information instead of the data itself. This is useful for dealing with privacy concerns and knowledge reuse [35]. 

The ensemble clustering algorithm consists of two main stages: diversity and consensus function. The data set is processed in the diversity stage with a single clustering algorithm with several initializations or multiple standard clustering algorithms. The results of this process are recorded-based clustering. Afterwards, the consensus function is implemented to combine the based clustering result and produce the final consensus solution. Currently, there is a different approach for consensus function, such as conducting co-association matrix or hypergraph partitioning. With a co-association matrix, a key advantage is the specification of a number of clusters in consensus partition is not required [30]. However, hypergraph partitioning requires this specification information [30]. However, in this research, the cluster number will be investigated and is used as input of hypergraph partitioning. As a result, a co-association matric will be unsuitable in this case. Thus, hypergraph partitioning is chosen for the consensus function in this research.

In the hypergraph, there are two main components: hyperedges present clusters, and vertices present equivalent samples or points. A clustering is presented as a label vector $\;\vartheta^{t}$. Label vector ϑ with mixture of r label vector $\;\vartheta^{1},\vartheta^{2},\;\ldots ,\vartheta^{r}$, where r is known as number of clustering, is the procedure by consensus function. The objective function is described by function $T\;:V^{v*r}\;\to V^{v}$ mapping a set of clustering to an integrated clustering $T\;:\{\vartheta^{t}\;|t\;\in \left\{1,2,\;\ldots ,\;r\right\}\;\to \;\vartheta \;\}.$ Labelled vector of $\vartheta^{t}$ is demonstrated by binary matrix $L^{t}$ where each cluster is specified with a column. In situation where the row is relating to an object with a known label, and all entries of the row in the binary membership indicator matrix $L^{t}$ are considered equal to 1. In contrast, in the situation where the row is relating to an unknown label, objects are considered equal to 0. Matrix $L=\left(L^{1}\;L^{2}\ldots \;L^{r}\right)$ as a hypergraph adjacency matrix is explained with v vertices and $a=\overset{r}{\underset{e=1}{\sum }}g^{e}$ hyperedges.

There are three algorithms in hypergraph methods: Cluster-based Similarity Partitioning Algorithm (CSPA), HyperGraph Partitioning Algorithm (HGPA) and Meta-Clustering Algorithm (MCLA).

In the Cluster-based Similarity Partitioning Algorithm, clustering can be used to generate a measure of pair-wise similarity because it illustrates the relationships among the objects that reside within the same cluster. The fraction of the clustering, where two objects occur within the same cluster and can be calculated in one sparse matrix multiplication $\frac{1}{r}\;L\;L^{c}$ where  L is indicator matrix and ${\;L}^{\;c}\;$ is matrix transposition, it is indicated by the entries of B [35]. The purpose of the similarity matrix is to re-cluster the items using any suitable similarity-based clustering technique.

HyperGraph Partitioning Algorithm (HGPA) partitions the hypergraph by cutting the smallest number of hyperedges possible. All hyperedges are weighed to ensure that they are all of the same weight. Furthermore, all vertices have the same weight. The partitions are created using the minimal cut technique, which divides the data into I unconnected components of roughly equal size. For these partitions, Han et al. (1997) employed the HMETIS hypergraph partitioning package [36]. In contrast to CSPA, which considers local piecewise similarity, HGPA solely considers the comparatively global links between items across partitions. Furthermore, HMETIS has a proclivity for obtaining a final partition in which all clusters are nearly the same size.

Cluster correspondence is dealt with in MCLA integration. MCLA finds and consolidates cluster groups, transforming them into meta-clusters. Constructing the meta-graph, computing meta-clusters, and computing clusters of the objects are the three key aspects of this method for finding the final clusters of items. The hyperedges $J^{e},\;e=1,2,\;\ldots ,\;a$ are the meta-vertices, and the graph and edge weights are proportional to the similarity between vertices. Matching labels can be identified by partitioning the meta-graph into o balanced meta-clusters. Each vertex is appropriately weighted to the size of the cluster to which it belongs. Balancing ensures that the sum of vertex-weights is generally equal within each meta-cluster. To achieve clustering of the J vectors, the graph partitioning package METIS is used in this stage. Each vertex in the meta-graph represents a distinct cluster label; as a result, a meta-cluster denotes a collection of corresponding labels. The hyperedges are crushed into a single meta-hyperedge for each of the o meta-clusters. An object is assigned to the meta-cluster with the highest entry in the association vector. Ties are broken in an ad hoc manner using this approach.

For this research, ensemble clustering is used to improve the robustness and quality of results from the K-means clustering algorithm. At the diversity stage, the K-means clustering algorithm with various initial choices of a centroid is applied to the data set to create base clustering. Following base clustering, in the consensus function stage, CSPA, HGPA, and MCLA algorithms are used separately to conduct the final results. Following base clustering, in the consensus function stage, the CSPA, HGPA, and MCLA algorithms are used separately to conduct the final results. The ROM of trunk, hip and knee as features were passed through K-means with squared Euclidean distance and random initial choice of centroid using Matlab multiple times (50 times) to form-based clustering for ensemble cluster. Before passing these results to ensemble clustering, duplicated results from K-means were removed to increase the quality and diversity of based clustering. From obtained-based clustering, ensemble clustering uses CSPA, HGPA, and MCLA as consensus functions processed to finalize the final result using the python ClusterEnsembles package.

### 2.6. Clustering—Wards

Besides partitioning clusters, the other widely utilized clustering algorithm is Hierarchical Clustering. The Hierarchical Clustering algorithm produces clusters in a hierarchical tree-like structure or a dendrogram [26,37]. 

In the beginning, each data point is assigned as a single unique cluster. By combining two data sets, allocating the data to an existing cluster, or merging two clusters after each loop, a new cluster can be formed [38]. The condition to create a new cluster is when the similarity or dissimilarity between every pair of objects (data or cluster) is found. Currently, there are four common kinds of linkage techniques, but research has shown that Ward’s linkage is the most effective technique for dealing with noisy data compared to the other three [26,38,39]. 

The linkage technique developed by Ward in 1963 uses the incremental sum of squares, which means the growth in the total within-cluster sum of squares as a consequence of merging two clusters [26]. The sum of squares of the distance between entire objects in the cluster and the cluster’s centroid is explained as the within-cluster sum of squares [38]. The sum of squares metric is equal to the distance metric $d_{AB}$, which is shown below:(2)$${d^{2}}_{EF}=\frac{2n_{E}n_{F}}{n_{E}+n_{F}\;}\|\bar{y_{E}}+\bar{y_{F}}\|{\;}^{2}$$
where the Euclidean distance is represented by || ||, the centroid of cluster *E* and *F* is represented by $y_{E}$ and $y_{F}$ respectively, and the number of elements in cluster *E* and *F* is represented by $n_{E}$ and $n_{F}$. In some research studies as references, Ward’s linkage does not contain the factor of 2 in Equation (2) when $n_{E}$ is multiplied by $n_{F}$. This factor’s main purpose is to ensure that the distance between two singleton clusters will be the same as the Euclidean distance.

To calculate the distance from cluster *D* to a new cluster *C*, cluster *C* is formed by merging clusters *E* and *F*, and the updated equation is shown as follows:(3)$${d^{2}}_{DC}=\frac{n_{E}+n_{D}}{n_{C}+n_{D}}{d^{2}}_{DE}+\frac{n_{F}+n_{D}}{n_{C}+n_{D}}{d^{2}}_{DF}-\frac{n_{D}}{n_{C}+n_{D}}{d^{2}}_{EF}$$
where the distance between cluster *D* and cluster *E* is represented by $d_{DE}$, the distance between cluster *D* and cluster *F* is represented by $d_{DF}$, the distance between cluster *F* and cluster *E* is represented by $d_{EF}$, the number of elements in clusters *E*, *F*, *C* and *D* are represented by $n_{E}$, $n_{F}$, $n_{C}$ and $n_{D}$.

After the hierarchical cluster is formed, a cut point is determined which can be at any position in the tree so that a full description of the clusters (final output) can be extracted [26]. In this study, the ROM of the trunk, hip and knee were passed through Ward clustering with Euclidean distance using the linkage and cluster function in Matlab.

### 2.7. Determining Optimal Number of Cluster

In partitioning clustering, for example, K-means clustering, the number of cluster *h* to be formed is defined and given, and choosing the correct optimal number of clusters for a single data set is challenging. This question, unfortunately, has no definitive answer. The method for determining similarity and the partitioning parameters define the ideal number of clusters, which is highly subjective. A basic and popular strategy is to examine the dendrogram produced by hierarchical clustering to see if it offers a specified number of clusters. Unfortunately, this strategy is as subjective. Direct technique and statistical testing method are two of these methods. Optimizing a criterion, such as the sum of squares within a cluster or the average silhouette, is a direct strategy. The equivalent procedures are known as the elbow and silhouette methods. The silhouette method analyses the average distance between clusters while the elbow method analyses the total within-cluster sum of square (WSS) for different clusters. In this research, both elbow and silhouette methods are used to determine optimal number of cluster for K-means cluster and Ward clustering using evalclusters function from Matlab and KElbowVisualizer function from Yellowbrick.

### 2.8. Machine Learning—Classification

A Bayesian neural network [15,40] construction operates a feed-forward structure with three layers, and it is formed by:(4)$$z_{k}\left(x,w\right)=f\left(b_{k}+\overset{l}{\underset{i=1}{\sum }}w_{ki}f\left(b_{i\;}+\overset{m}{\underset{j=1}{\sum }}w_{ij}x_{j}\right)\right)$$
where the transfer function is represented by f(.), and in this paper, the hyperbolic tangent function is applied, the number of input nodes is represented by m (j starts from 1 to m), the number of hidden nodes is represented by l (i starts from 1 to l), the quantity of output is represented by q (k starts from 1 to q), the weight from input unit $x_{j}$ to the hidden unit $y_{i}$ is represented by $w_{ij}$, the weight from hidden unit $y_{i}$ to the output $z_{k}$ is represented by $w_{ki}$, and biases are represented by $b_{i\;}$ and $b_{k}$.

Bayesian regularization structure is suggested to improve the generalization capabilities of the neural network irrespective of whether the presented data are noisy and/or finite [41]. In Bayesian learning, the probability distribution of network factors will be observed; thus, the trained network’s greatest generalization can be delivered. Particularly, all of the obtainable data can be compatible and used in this kind of neural network to train. Consequently, the application with a small data set is appropriate.

The best possible model in the Bayesian framework, which the training data S corresponded to, is acquired automatically. Founded on Gaussian probability distribution on weight values, applying Bayes’ theorem can compute the posterior distribution of the weights w in the network H and this is presented below:(5)$$p(w\;|S,H)=\frac{p(S\;|w,H)p(w|H)}{p\left(S|H\right)}\;$$
where p(S |w,H) represents the probability that knowledge about the weight from observation is included, the knowledge about background weight set is contained in the prior distribution  p(w|H), and lastly, the p(S|H) represents the network H evidence.

For a MLP neural network described in Figure 3, the cost function  G(w) can be minimized to achieve the most possible value for the neural network weight $w^{MP}$, and the cost function is shown below:(6)$$G\left(w\right)=\beta K_{S}\left(w\right)+\;\alpha K_{W}\left(w\right)\;$$
where hyper-parameters are symbolized by α and β, and the effective difficulty of network structure is operated by the ratio  α/β, the error function is symbolized by $K_{S}\left(w\right)\;$ and the total square of weight function is symbolized by $K_{W}\left(w\right)$; this function can be calculated using the below equation:(7)$$K_{W}\left(w\right)=\frac{1}{2}{\left\|w\right\|}^{2}$$

Updating the cost function with hyper-parameters, the neural network with a too large weight can lead to poor generalization when new test cases are used and can be avoided. Consequently, during a neural network training process, a set of validation is not compulsory.

To update hyper-parameters, the Bayesian regularization algorithm is used, and it is shown below:(8)$$\alpha^{MP}=\frac{\gamma }{2K_{W}\left(w^{MP}\right)};\;\beta^{MP}=\frac{N-\gamma }{2K_{S}\left(w^{MP}\right)}\;$$
where the effective number of parameters is represented by $\gamma =c-2\alpha^{MP}tr{\left(H^{MP}\right)}^{-1}$, the total number of parameters in the network is represented by c, the total number of errors is represented by N, and the Hessian matrix of G(w) at the smallest, minimum point of $w^{MP}$ is represented by H. 

The Bayesian framework can estimate and evaluate the log evidence of model $H_{i}$, and it is shown below:(9)$$\ln p\left(S|H_{i}\right)=-\alpha_{MP}K_{W}^{MP}-\beta_{MP}K_{W}^{MP}-\frac{1}{2}\ln\left|A\right|+\frac{W}{2}\ln\alpha_{MP}+\;\frac{N}{2}\ln\beta_{MP}+\ln M+2\ln M+\frac{1}{2}\ln\frac{2}{\gamma }+\frac{1}{2}\ln\frac{2}{N-\gamma }\;\;$$
where the number of network parameters is represented by W, the number of hidden nodes is represented by M, and the cost function Hessian matrix is represented by A. The best optimal structure will be found out based on the log evidence value; a structure which has the highest value will be chosen.

To measure multi-class classification performance, the familiar performance metric can be considered and used: precision, recall and accuracy. These indicators are shown as follows:(10)$$Recall=\frac{TP_{O}}{TP_{O}+FN_{O}}$$
(11)$$Precision=\frac{TP_{O}}{TP_{O}+FP_{O}}$$
(12)$$Accuracy=\frac{Total\;of\;correctly\;classifed\;data}{Total\;number\;of\;data}$$
where the number of the data inputs, which is denoted as  O  and is classified correctly, is represented as $TP_{O}$ (true positive of O), and O denotes one of the classes in multi-class. The number of the data inputs, which does not denote as O and is classified as O, is represented as $FP_{O}$ (false positive of O), and the number of the data inputs, which denotes as O and is classified as not O, is represented as $FN_{O}$ (false negative of O).

The clustering algorithm’s output is combined with the original input data set to create a new Bayesian neural network classification data set. For the Bayesian neural network classification, the data set is separated into two sets: the first set contains 50% of the overall sets used for training purposes and the second sets has the remaining sets for testing purposes. In addition, to train this neural network classifier, the Levenberg–Marquardt with Bayesian regularization algorithm is implemented, and the mean squared error function is selected as the error function $K_{D}\left(w\right)$ [41]. In this study, cluster result (Wards Clustering, combination of K-means and CPSA, HGPA and MCLA) and features of the clustering algorithm’s data (ROM of trunk, hip and knee) were passed to the Bayesian neural network with maximum number of epochs of 600 and maximum mu of 1 × 10^100^. Minimum performance gradient is 1 × 10^−20^ using Matlab script. Fifty percent of the data set (239 trials) was used for training purposes, and the other half (234 trials) was used for testing purposes.

### 2.9. Statistical Analysis

Besides using the Bayesian neural network to choose the better unsupervised machine learning algorithm for classifying the lifting technique, the Partial ${Ƞ}^{2}$ (partial eta squared) is used to measure the effect size of different algorithms from the statistical point of view. 

A one-way multivariate analysis of variance (MANOVA) was used to examine the trunk, hip, and knee ROM differences between each cluster. Tukey’s Honestly Significant Difference test was conducted to analyse significant group differences. The statistical analysis methods (least significant difference) were performed with patient-reported outcome measures such as the pain self-efficacy questionnaire (PSEQ). All analyses were conducted with a significance level set at 0.05 using IBM SPSS software version 28.0.1 (SPSS Inc., Chicago, IL, USA).

## 3. Results

Four-hundred and seventy-three lifting trials were included in this study. The participants’ demographic information is summarized in Table 1.

The results of elbow and silhouette methods for the Ward clustering algorithm are shown in Figure 4 and Figure 5. The results of the elbow and silhouette methods for the K-means clustering algorithm are shown in Figure 6 and Figure 7. The elbow method for both Ward and K-means suggests the optimal number of clusters is two, while the silhouette method suggests that four is the optimal number of clusters. In this study, the cluster result represents the lifting technique that people with CLBP uses. Currently, lifting techniques can be classified as two techniques: a stoop lift or leg lift. The main object of this research is to identify how many possible lifting techniques people with CLBP can perform besides the two lifting techniques. As a result, the optimal number of clusters for both K-means and Ward clustering is four.

The descriptive statistics pertaining to the range of motion of the trunk, hip and knee for each cluster between different unsupervised machine learning methods are summarized in Table 2. 

Classification of four clusters by applying the Bayesian neural network as a classifier on the test set of different methods is summarized in Table 3. Additionally, the effect sizes of different unsupervised machine learning are shown in Table 4. For trunk, hip and knee features, Ward clustering provided the highest effect compared to the combination of K-means and ensemble clustering. The Bayesian neural network and effect size calculation result suggests that Ward clustering is the optimal algorithm for this study. 

The optimum number of hidden nodes of the Bayesian neural network training versus log evidence is plotted and represented in Figure 8. Based on the plot, the training model with nine hidden nodes is determined as the best classification indication.

Cluster 2 was the most common, which constituted 40.80% of all self-selected techniques. The least chosen was Cluster 1 (6.76% of all data analysed). The descriptive statistics pertaining to PSEQ and pain level for each cluster are summarized in Table 5. 

The result of the hierarchical tree is represented using a dendrogram shown in Figure 9. Four clusters were created. A 3D scatter plot was generated to visualize the clustering algorithm’s output in three-dimensional space, and it is shown in Figure 10.

There were significant differences between all clusters and body regions (*F* (9, 1136) = 195.67, *p* < 0.0001). The post hoc test revealed different features between the clusters summarized in Figure 11. Post hoc test results show that there were significant differences between all clusters for trunk, hip and knee ROM. This shows that the four cluster results are completely distinctive from a statistical point of view.

Clusters 2 and 4 are hip dominance with more knee movement than the trunk. Cluster 3 is hip dominance with more trunk movement compared to the knee. Cluster 1 is knee dominance with more hip movement than the trunk.

Post hoc test revealed PSEQ between the clusters, summarized in Figure 12. The PSEQ mean scores of cluster 1 is statistically significantly different from other clusters (*p* < 0.05). The PSEQ mean scores are not statistically significantly different between clusters 2, 3, and 4 (*p* > 0.05 for each pair of clusters). From a clinical point of view, if the PSEQ is greater than 40, there is minimal impairment, and the patient is very confident. If the PSEQ value is less than 40, it means there is an impairment, and the patient is not too confident. As a result, it makes only cluster 1 (PSEQ mean score is 36.56) different from the other clusters.

## 4. Discussion

To the best of our knowledge, this is the first study to utilize unsupervised and supervised machine learning algorithms to classify different movement strategies during a dynamic task using motion capture camera. Besides stoop lift (represent as cluster 3) and leg lift (represent as cluster 1), the study found two additional lifting techniques in people with CLBP, which is in agreement with our study hypothesis. In clusters 2 and 4, people with CLBP tend to use the hip mainly with additional support from knee to lift. Although the Ward clustering method is different from the combination of K-means and ensemble clustering method, descriptive statistics of the trunk, hip and knee ROM results for each cluster between methods are similar. Both the Ward clustering method and the combination of K-means and ensemble clustering method suggest that there would be four different lifting techniques: hip dominance with more trunk movement (cluster 3—ward clustering, cluster 3—K-means and CSPA, cluster 1—K-means and HGPA, and cluster 3—K-means and MCLA); knee dominance with more hip movement (cluster 1—ward clustering, cluster 2—K-means and CSPA, cluster 4—K-means and HGPA, and cluster 4—K-means and MCLA); hip dominance with more knee movement than the trunk (cluster 4—ward clustering, cluster 4—K-means and CSPA, cluster 2—K-means and HGPA, and cluster 1—K-means and MCLA); and hip dominance with much more knee movement than the trunk (cluster 2—ward clustering, cluster 1—K-means and CSPA, cluster 3—K-means and HGPA, and cluster 2—K-means and MCLA). Additionally, the study suggested that Ward clustering was the optimal clustering method for the current data set.

The algorithm in this study resulted in high recall and precision values for all lifting clusters. These indicate that most of the data have been classified correctly, and the model performs well. There is potential to classify any new data using this result. In previous research, hierarchical clustering techniques have been applied in psychiatry. Paykel and Rassaby [42] applied hierarchical clustering to classify suicide attempters. The result has helped to investigate the causes and to guide some improvement for the method of therapy. Additionally, Kurz et al. [43] used Ward’s hierarchical agglomerative method to classify suicidal behaviour. This study’s accuracy is 95.7% (Ward’s method had classified 96% of all cases) [43]. Their clusters were found to have implications for clinical interpretation, therapy and prognostication. This demonstrates that hierarchical clustering can successfully provide some valuable information for further application. 

In this study, there are a few variables and aspects that might impact lifting technique in people with CLBP such as height, weight, age, duration of pain and lumbar muscle strength. However, multivariant analysis of covariance (MANCOVA) was conducted as further analysis to determine whether trunk, hip and knee ROM performances differed between each cluster whilst controlling for these variables. MANCOVA results indicated that there is a statistically significant difference between clusters in terms of combined trunk, hip and knee ROM, after controlling for height, weight, age, duration of pain and lumbar muscle strength. Therefore, the results from MANCOVA suggested that there was no significant differences in our initial MANOVA analysis. 

Lifting is an important risk factor associated with work-related CLBP [44]. People with CLBP utilized different lifting techniques compared to people without CLBP, particularly, less trunk ROM and increased knee ROM, which are identified as one of many phenotypes of lifting techniques in this study [45]. Identification of lifting techniques in people with CLBP is potentially important in rehabilitation. Physiotherapists and manual handling advisors often encourage CLBP patients to lift with a straight back, which is already the most common lifting technique identified in this study [46]. It is unclear whether the straight-back lifting technique is the cause or the result of the motor adaptation of CLBP. Therefore, it is perhaps unsurprising that the general advice to keep the back straight during lifting did not prevent future low back pain [46]. Identifying lifting techniques and their impact on the lumbar spine can help clinicians to guide LBP patients in adopting a lifting technique that imposes the least amount of force on the lumbar spine. This information can help clinicians in directing effective ergonomic interventions in people with occupational LBP. Additionally, identifying lifting techniques in people with CLBP may help physiotherapists direct and prioritize rehabilitation towards appropriate areas of the body that may be associated with a painful lifting technique (e.g., trunk, hip, and knee). This may result in positive changes in pain and CLBP-related disability (i.e., precision medicine). 

An important finding of this study is that despite the majority of the clusters demonstrating similar pain levels, cluster 1 demonstrated the lowest pain self-efficacy, which was associated with the least amount of trunk movement and the most knee movement during lifting (i.e., lifting with a ‘straight’ back). This finding is consistent with the previous literature that pain self-efficacy is positively correlated with lifting lumbar ROM in people with CLBP [47]. The lower self-efficacy in this group may manifest as reluctance to bend the lumbar spine during lifting, which in a past study has been associated with higher lumbar extension load [48]. This in turn, may sensitize lumbar tissues resulting in pain perpetuation. Thus, observation of a patient’s lifting technique and pain self-efficacy may be key clinical assessments to define this group and develop appropriate multicomponent interventions, such as education and exercise [49]. 

One should interpret these study results with caution. One limitation of this study is the small sample size. The reader needs to be cautious when applying the result to a larger sample size. However, in previous studies with an even smaller sample size (*n* = 236 [42] and 486 [43]), this method could still cluster participants accurately. Another limitation of this study is the clustering performed on the repetitions instead of mean joint angles for each participant, as this study aims to identify and capture as many different lifting techniques that people with CLBP could demonstrate. This means that we could not account for within-participant lifting technique variability (i.e., variation between each lifting repetition performed by a single participant). Pain alters movement variability within and between participants due to differences in muscle recruitment strategies, psychological features (e.g., fear-avoidant behaviour) and muscle function (e.g., strength, flexibility) to name a few [50,51]. As a result, a small number of CLBP participants in this study (*n* = 17 or 15% of total participants) demonstrated employed >1 lifting technique (i.e., belonged to >1 lifting clusters). The clinical implication of this limitation is currently unclear and should be evaluated in future studies. Third, the experiment only focused on the sagittal plane’s trunk, hip and knee ROM. As such, we were unable to capture movement of the trunk, hip and knee in the other coronal and axial planes. However, the symmetrical lifting task involves mainly movements of the lumbar spine, hip and knee in the sagittal plane. Future studies should explore clustering techniques involving movements in all planes.

It is currently unknown if lifting technique clusters in healthy people are different from those with CLBP. Future studies should aim to compare different lifting techniques in healthy people and people with CLBP. This information will guide the assessment and rehabilitation of movement in people with CLBP. A validation of this novel methodology from a clinic point of view can be conducted in future studies. Authors should discuss the results and how they can be interpreted from the perspective of previous studies and of the working hypotheses. The findings and their implications should be discussed in the broadest context possible. Future research directions may also be highlighted.

## 5. Conclusions

To the best of our knowledge, this research is the first study introducing an innovative methodology for classifying different movement strategies during lifting tasks in people with CLBP using unsupervised and supervised machine learning techniques. The optimal unsupervised machine learning technique based on Ward’s clustering accurately differentiated four distinct movement groups in people with CLBP instead of two lifting techniques, as in current state-of-the-art research studies. The output of the clustering (four clusters) has been validated by the supervised machine learning using Bayesian Neural Network with an accuracy of 97.9%. This promising technique could aid in more precise assessment and rehabilitation of people with CLBP.

## Figures and Tables

**Figure 1 sensors-22-06694-f001:**
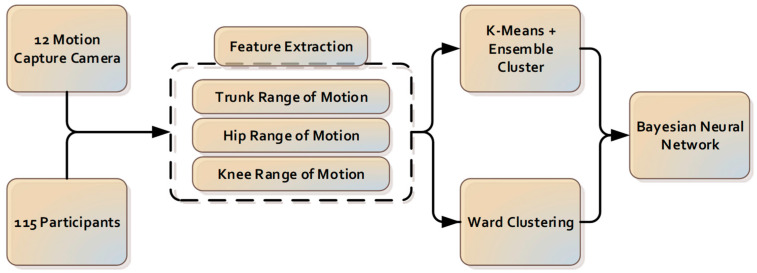
The components for the camera-based cluster classification system.

**Figure 2 sensors-22-06694-f002:**
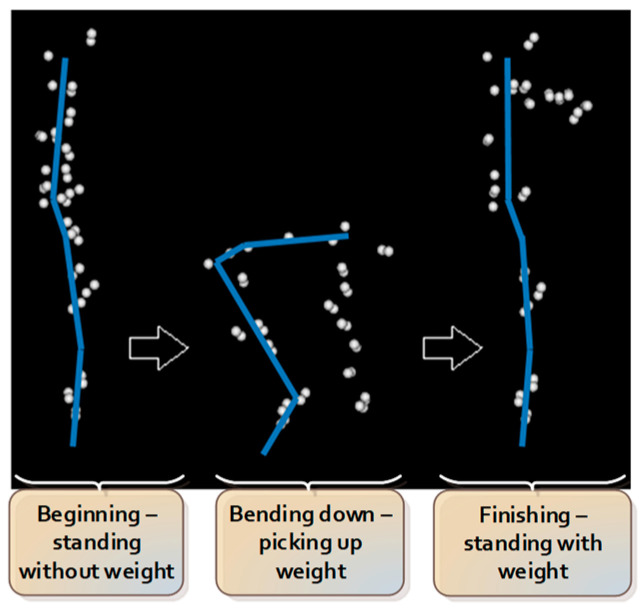
Sequence of consecutive actions during lifting.

**Figure 3 sensors-22-06694-f003:**
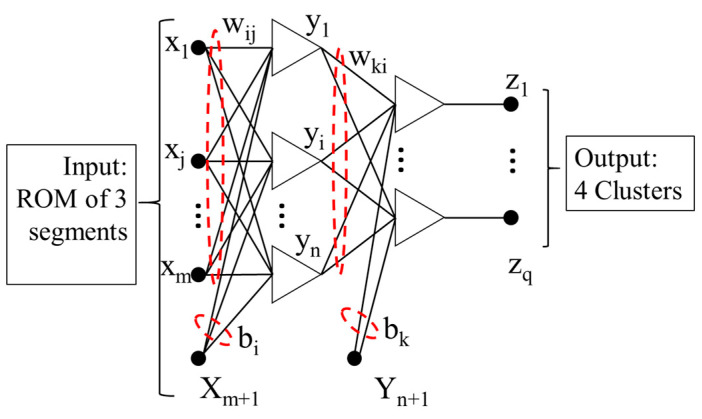
Artificial neural network (ANN) structure (weights and biases are highlighted in red dotted circle).

**Figure 4 sensors-22-06694-f004:**
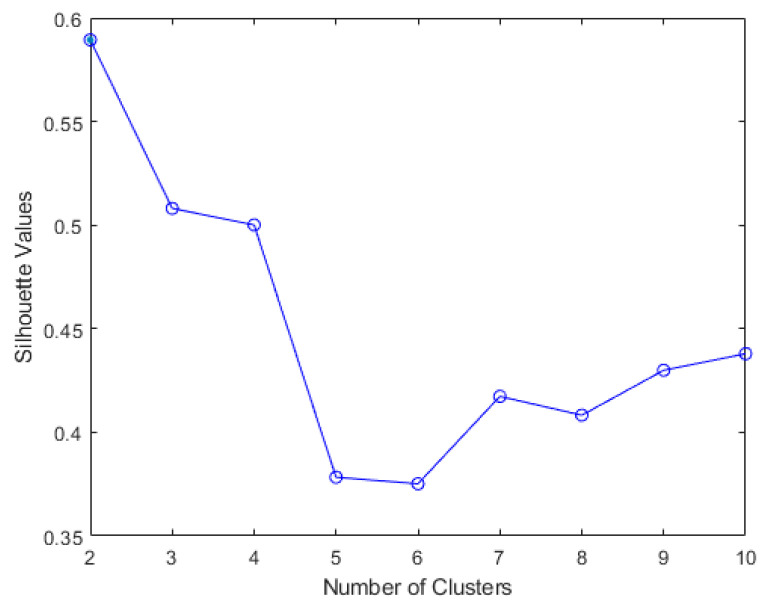
Silhouette score results for Ward clustering algorithm.

**Figure 5 sensors-22-06694-f005:**
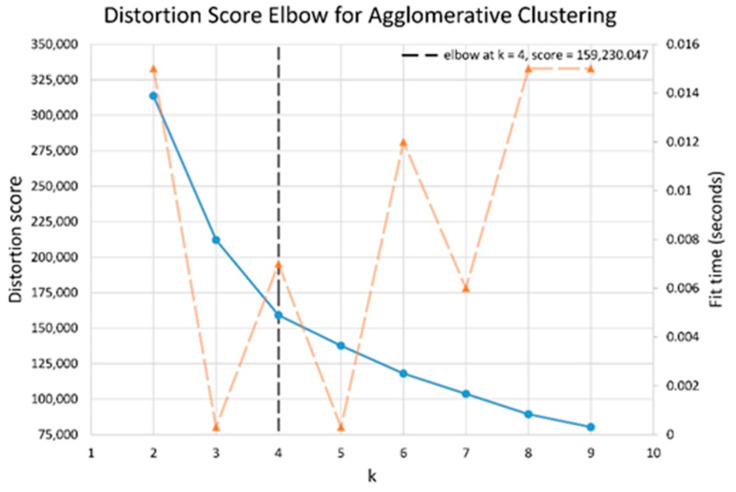
Elbow method results for Ward clustering algorithm (blue line indicates distortion score, and orange dashed line indicates the time to train the clustering model).

**Figure 6 sensors-22-06694-f006:**
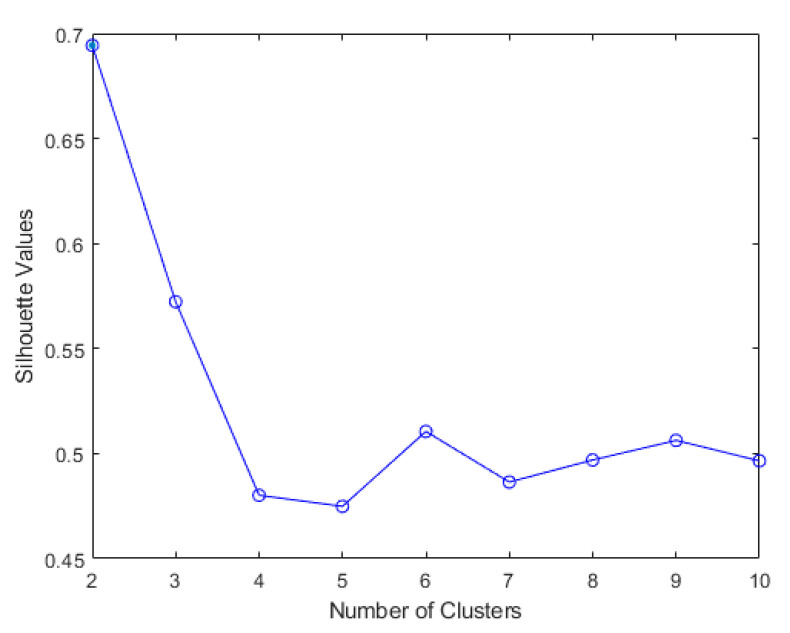
Silhouette score results for K-means clustering algorithm.

**Figure 7 sensors-22-06694-f007:**
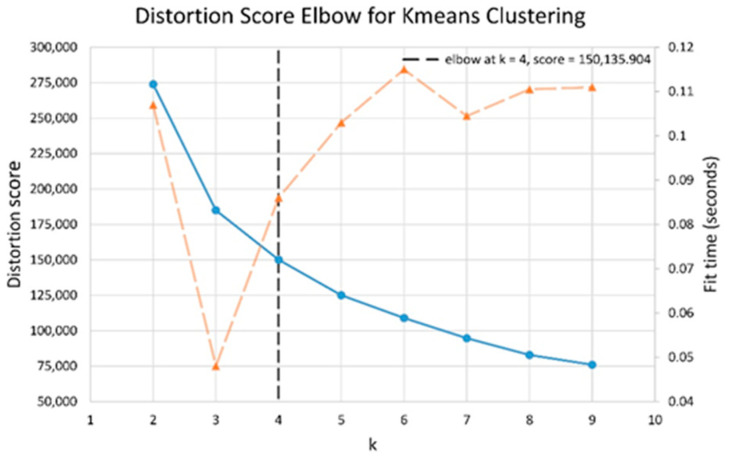
Elbow method results for K-means clustering algorithm (blue line indicates distortion score, and orange dashed line indicates the time to train the clustering model).

**Figure 8 sensors-22-06694-f008:**
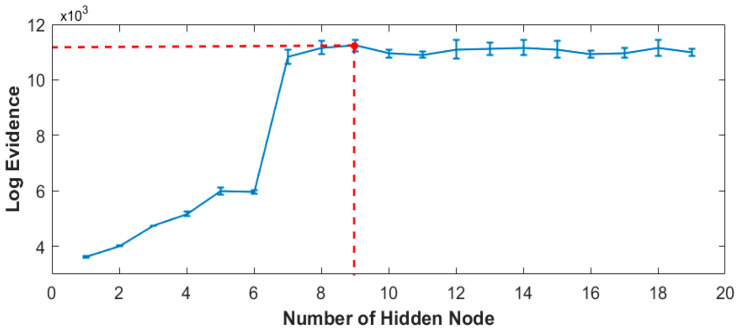
Evidence framework of Bayesian inference (optimum number of hidden node of 9, indicated by red dashed line).

**Figure 9 sensors-22-06694-f009:**
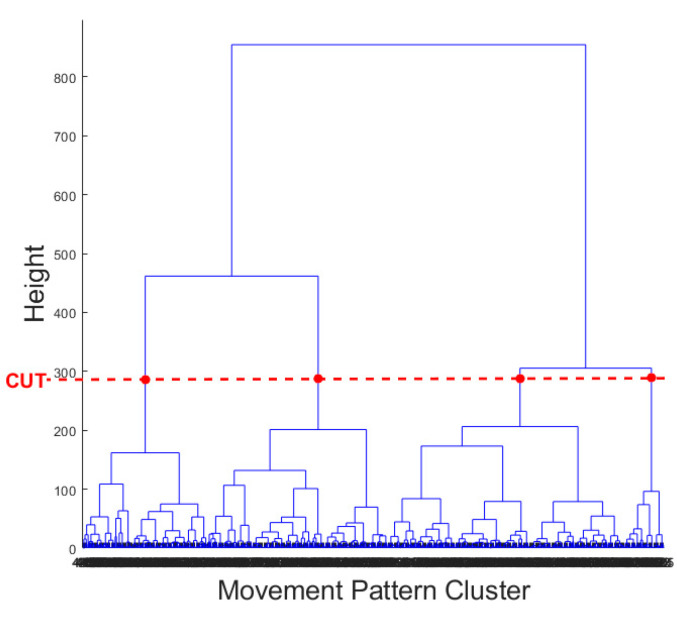
Dendrogram of output hierarchical tree (red dashed line indicates a cut point to form four clusters).

**Figure 10 sensors-22-06694-f010:**
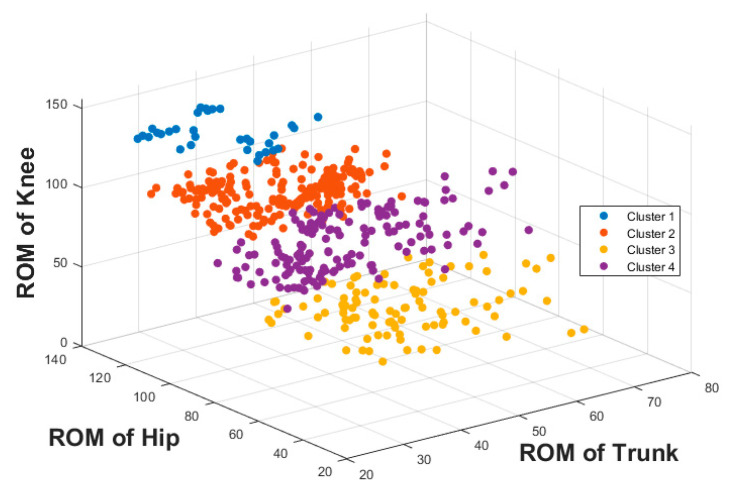
Three-dimensional scatter plot of output of cluster algorithm.

**Figure 11 sensors-22-06694-f011:**
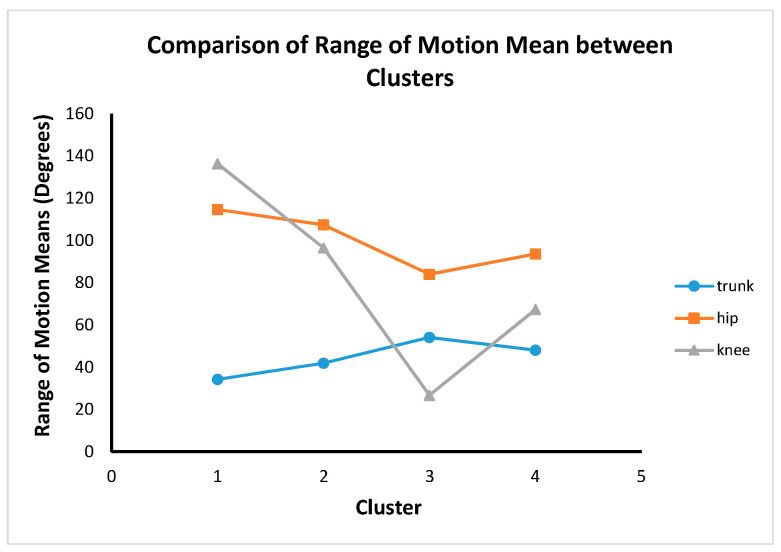
Estimated marginal means of trunk, hip and knee ROM between clusters.

**Figure 12 sensors-22-06694-f012:**
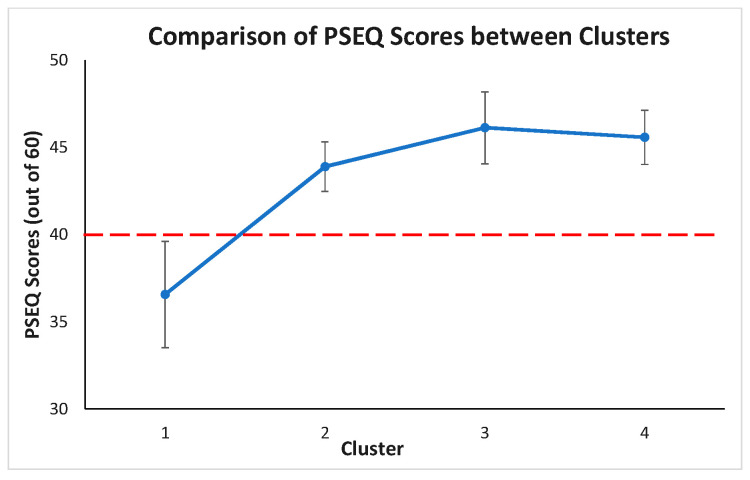
Estimated marginal means of PSEQ between clusters (red dashed line indicates threshold value for PSEQ; below red line suggests clinically significant finding or low pain self-efficacy).

**Table 1 sensors-22-06694-t001:** Descriptive participants’ demographic information.

Variables (Units)	Mean (SD)
Age (years)	45.4 (11.6)
Height (cm)	173.4 (11.1)
Weight (kg)	79.6 (17.6)
BMI (m/kg^2^)	26.3 (5.4)
Pain Level (VAS out of 100)	45.8 (19.9)
Duration of Pain (months)	89.2 (113.3)
ODI (%)	31.5 (14.4)
PSEQ (out of 60)	45.2 (9.9)

BMI, body mass index; ODI, Oswestry Disability Index; PSEQ, Pain self-efficacy questionnaire; VAS, Visual Analogue Scale; SD, standard deviation.

**Table 2 sensors-22-06694-t002:** Descriptive statistics (mean (SD)) of trunk, hip and knee range of motion for each cluster between methods.

		Mean (Standard Deviation)		
		Trunk	Hip	Knee
Ward Clustering	Cluster 1	34.19 (6.84)	114.58 (9.74)	136.24 (12.35)
	Cluster 2	41.88 (7.84)	107.37 (9.27)	96.39 (12.66)
	Cluster 3	54.00 (9.26)	83.92 (13.02)	26.70 (10.36)
	Cluster 4	48.04 (9.73)	93.54 (9.61)	67.29 (13.72)
K-means + CSPA	Cluster 1	34.19 (6.84)	105.20 (9.67)	88.75 (7.46)
	Cluster 2	33.01 (7.35)	108.60 (11.55)	115.13 (15.31)
	Cluster 3	53.17 (9.09)	86.44 (13.84)	31.72 (16.95)
	Cluster 4	47.28 (9.82)	93.91 (10.35)	64.57 (10.53)
K-means + HGPA	Cluster 1	52.80 (9.89)	84.26 (13.06)	30.92 (14.99)
	Cluster 2	47.65 (9.24)	96.07 (9.26)	65.36 (10.90)
	Cluster 3	44.92 (8.93)	105.23 (9.61)	89.02 (7.81)
	Cluster 4	38.11 (7.38)	108.59 (11.62)	115.08 (15.45)
K-means + MCLA	Cluster 1	47.52 (9.52)	95.24 (10.68)	65.59 (11.18)
	Cluster 2	42.58 (8.59)	105.79 (10.47)	95.49 (10.07)
	Cluster 3	53.90 (9.16)	84.23 (12.98)	27.18 (10.60)
	Cluster 4	36.09 (7.05)	111.80 (9.95)	129.71(13.84)

**Table 3 sensors-22-06694-t003:** Classification results (recall, precision and accuracy) of Bayesian neural network between each cluster on test set.

Method	Cluster	Recall	Precision	Accuracy
**Ward Clustering**	Cluster 1	93.8%	93.8%	**97.9%**
	Cluster 2	99.0%	96.9%	
	Cluster 3	98.0%	100%	
	Cluster 4	97.2%	98.6%	
K-means + CSPA	Cluster 1	93.1%	88.5%	93.6%
	Cluster 2	88.1%	94.5%	
	Cluster 3	94.9%	98.2%	
	Cluster 4	98.3%	93.5%	
K-means + HGPA	Cluster 1	98.3%	96.7%	94.9%
	Cluster 2	89.8%	98.1%	
	Cluster 3	96.6%	89.1%	
	Cluster 4	94.9%	96.6%	
K-means + MCLA	Cluster 1	94.2%	98.5%	97.0%
	Cluster 2	100%	94.8%	
	Cluster 3	98.1%	98.1%	
	Cluster 4	91.7%	100%	

**Table 4 sensors-22-06694-t004:** Partial eta squared effect size result between different methods.

Method	Trunk	Hip	Knee
**Ward Clustering**	**0.015**	**0.078**	**0.122**
K-means + CSPA	0.002	0.013	0.029
K-means + HGPA	0.007	0.010	0.043
K-means + MCLA	0.001	0.003	0.058

**Table 5 sensors-22-06694-t005:** Descriptive statistics (mean (SD)) of PSEQ, and pain level for each cluster.

		Cluster 1	Cluster 2	Cluster 3	Cluster 4
PSEQ	Mean	35.56	43.89	46.13	45.57
	(SD)	(8.45)	(9.95)	(10.53)	(9.5)
Pain	Mean	51.03	50.33	47.73	48.17
	(SD)	(10.22)	(20.40)	(21.21)	(19.27)

SD, standard deviation.

## Data Availability

For third-party data, restrictions apply to the availability of these data. Data were obtained from the University of Melbourne and are available from Pranata, A. et al. or at https://www.sciencedirect.com/science/article/pii/S0021929018301131 (accessed on 26 July 2020) with the permission of the University of Melbourne.

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
