# Peer review of "Machine Learning Derived Lifting Techniques and Pain Self-Efficacy in People with Chronic Low Back Pain"

_sensors, 2022, doi:10.3390/s22176694_

Round 1
Reviewer 1 Report
Authors proposed a novel approach, based on unsupervised and supervised machine learning algorithms, to cluster different strategies of patients with Chronic Low Back Pain. The study is well written, the results support the hypotheses, and the methodology is innovative and could be transferred to other applications.
Following are some suggestions that I think may improve your work.
Line 125: please specify the number of months along which the pain lasted
Line 133: May the different heights of participants, who must take a box of the same dimension, influence the strategy? (i.e. taller participants have to bend more).
Line 133: May the different strengths of participants, who must take a box of the same weight, influence the strategy? (i.e. weaker participants require a stronger effort).
Line 234: please indicate what r, L, and Lc mean in the equation
Line 283-284: Maybe there are some problems in the conversion of your document to pdf, because I cannot properly see the equations
Line 311: may you briefly describe the elbow and the silhouette methods, or eventually provide the name of the Matlab function you used
Line 315: Recently, a novel approach was proposed to separate biological data into different clusters through a Bayesian approach (Mezzetti et al. A Bayesian approach to model individual differences and to partition individuals: case studies in growth and learning curves. Stat Methods Appl (2022). https://doi.org/10.1007/s10260-022-00625-6). I suggest to cite this work because in line with the Bayesian approach.
Table 1: what does the mean and std mean for gender? Don’t you need a single value identifying the fraction of female participants?
Line 534: you affirmed that yours ‘is the first study to utilize unsupervised and supervised machine learning algorithms to classify lifting techniques in people with CLBP’. Are there studies investigating other dynamic tasks that use a similar approach? If so please cite them, otherwise emphasize the novelty of your approach with a statement such as: ‘By the best of our knowledge, our is the first study to utilize unsupervised and supervised machine learning algorithms to classify different movement strategies during a dynamic task’. (similar point in conclusions).
Is the clustering performed on the repetitions or on the participants? Since it looks on the repetition, does the same participant exploits the same strategy along all the repetitions? Why not clustering the mean joint angles (averaged along repetitions) of participants?
What do you expect that may influence the strategy exploited by a participant? In other words, why different participants have different strategies? Did you find a correlation with the lasting of the pain? Did you recorded whether participants had some physiotherapic threatment to solve their problem? I suggest, at least, to add some points in the discussion about the source of this inter-subject variability, and eventually check if there is a relation between the strategy and the lasting of the pain, the height, weight, or age.
Reviewer 2 Report
For the authors:
thank you for the submission of your manuscript in sensor. I have thoroughly read your study. It has been a great pleasure to review this study.
This is an interesting manuscript regarding a mixed machine learning approach to assess different lifting techniques in subjects with chronic low back pain.
I believe this work has some merit and touches novel aspects. Some adjustments are surely required.
Abstract and keywords
- Please explicitly report the abbreviation CLBP also in the abstract
- Please sensibly reduce the number of keywords!
Introduction
- Too long. Please shorten the sections reporting well-known aspects of machine learning. Just report those that are crucial to your study purpose.
- Line 84-85: this sentence is not needed here
- Line 88-95: this paragraph is not needed here
- Line 96-100: this paragraph is not needed at all
Methods
- Line 103-114: this paragraph is not needed. Please incorporate it into the others.
- Figure 1: please enrich this figure, it can become very informative (and summarize the entire paragraph): how many subjects? how many cameras? how many/which features extracted? which clustering/classification method?
- Line 120-129: this section can be move to a stand-alone sub-chapter called “participants”
- Line 125: “for >months”. Something missing here
- When describing the clustering methods, the description goes back and forth and there’s some confusion. Please shorten the descriptions if they’re well-known aspects (like K-means), give the same importance to the clustering methods (you start talking about partitional and hierarchical, but you then talk of partitional only and in a subsequent paragraph mention the hierarchical…this should be all clarified and smoothed).
- Line 211- 214: why did you state this here? Is it an assumption? Is there a referenced reason why?
Results
- Also here, there’s a bit of back-and-forth descriptions. Some methodological aspects are reported in the results instead of the methods. Please adjust
- Table 1: please use a number of decimals adequate to your measurement accuracy. Usually, one decimal is enough here.
- Line 421-430: please shrink this paragraph and make the choice between 2 and 4 clusters clearer.
- Line 456-461: as previously stated, this is an example of what should be stated in the methodology and not here
- Figure 8: why 8 clusters?
- Figure 9: does “leg” means “knee” here?
- Figure 11: nice figure. But your data refer to the mean ROM. Here, you use these ROM data as mean absolute values. This is not correct. The fact that, e.g., Cluster 1 knee ROM is 136° doesn’t mean that you can report that subject at 136° of knee flexion. This figure should be removed. Figure 12 is less impactful but has sense to describe the 4 clusters.
Discussion
- Discuss your main finding first.
- Put a larger accent onto the practical implications of this study.
- Limitation section is ok
